# Influencing Factors for Expatriation Willingness of Chinese Medical Aid Team Members (CMATMs) in Africa: A Qualitative Descriptive Study

**DOI:** 10.3390/ijerph17020632

**Published:** 2020-01-18

**Authors:** Xiaochang Chen, Xiaojun Liu, Zongfu Mao

**Affiliations:** 1School of Humanities and Management, Guangdong Medical University, 1# Xincheng Road, Dongguan 523808, China; 2Institute of Health Law and Policy, Guangdong Medical University, 1# Xincheng Road, Dongguan 523808, China; 3School of Health Sciences, Wuhan University, 115# Donghu Road, Wuhan 430071, China; xiaojunliu@whu.edu.cn; 4Global Health Institute, Wuhan University, 115# Donghu Road, Wuhan 430071, China

**Keywords:** expatriate, expatriation willingness, Chinese medical aid team, China, Africa

## Abstract

Chinese medical aid team members (CMATMs) play an important role in the implementation of China’s health assistance strategies in Africa. This paper explored the influencing factors of expatriation willingness for Chinese medical aid team members (CMATMs). We employed a qualitative descriptive study using semi-structured interviews with twenty-five participants. Participants included hospital directors and local Health and Family Planning Commission (HFPC) officers who were in charge of CMATMs dispatching, and CMATMs that had returned from medical aid service. Six influencing factors emerged: career advancement, loneliness, living conditions, personal safety, family–work conflict, and doctor–patient relationship. Career advancement is the most important factor and concern for doctor CMATMs. Social use of Internet is on the core of entertainment. Enhancing technical title promotion policies is the most important motivator. This study obtained baseline information that is useful to relevant stakeholders in their attempts to improve CMATMs’ expatriation willingness.

## 1. Introduction

Ever since dispatching medical aid teams to Africa in 1963, China has been showing its strong willingness to participate in global governance as well as to help other developing countries, especially African countries. Offering health assistance is one of China’s priorities in providing assistance to African countries, and one of the main channels is dispatching medical aid teams to these countries. By 2017, China had sent 25,000 medical aid team members to African countries [1], and there are currently still thousands of Chinese medical aid team members (CMATMs) providing services in recipient countries [1,2]. Obviously, CMATMs have a significant impact on the implementation of China’s health assistance strategies in Africa. However, some studies have declared that China is facing a dilemma between the increasing demand for health assistance and the difficulty of dispatching medical aid personnel to African countries, as medical staff have shown low willingness to be dispatched to African countries as CMATMs [3,4]. Expatriation willingness—that is, candidate’s willingness to accept expatriate assignment—is an important predictor for expatriate success [5]. Considering the significant role of CMATMs, needless to say, it is necessary and important to explore the influencing factors for expatriation willingness of CMATMs, because it is the premise to understand and boost the expatriation willingness of this population. However, little has been determined so far regarding this particular population, let alone the expatriation willingness and its influencing factors of this population. Therefore, the present study aimed to identify the influencing factors for expatriation willingness of CMATMs in Africa, and to provide scientific evidence for implementing relevant measures to enhance expatriation willingness of this particular population.

The work–family conflict is among the most discussed factors for discouraging expatriation willingness. Family issues are often cited as a main reason for refusing international assignments [6,7,8,9]. Studies focused on long-term expatriation have found that married candidates may resist and refuse overseas expatriation for the spouse’s reluctance to move or/and for the quality of education available to children in the host countries [8,9]. It is understandable that since long-term expatriation lasts more than one year, expatriates probably will take their family along, hence having to consider spouse willingness to move and the education quality for kids. Whereas, studies have found that short-term expatriates are usually not accompanied by their family [10]. However, it should be pointed out that although the expatriation period of CMATMs generally lasts for two years, which is considered as long-term expatriation, CMATMs are hardly accompanied by their family. Hence, while the prior studies focusing on long-term expatriation are informative, some of their contexts are different from that of CMATMs, hence the previous findings may not apply to this study. One significant difference is the children education issue. Since CMATMs do not take their children to the recipient country, the education issue in the recipient country is definitely not an influencing factor for the expatriation willingness of CMATMs. However, since CMATMs must stay in the recipient country for 2 years, which means they could not take care of their family during this period, it is the same case with the spouse affair, children care affair, and eldercare affair for CMATMs. Overall, CMATMs will not be able to fulfill their family roles, which include their role as spouse, parent, and offspring. 

Living conditions in the host country have also been found related to expatriation willingness. Employees from advanced regions are most reluctant to relocate to developing economies due to the living conditions [11]. Prior studies have suggested that countries with a high level of economic level are favored by expatriate candidates, especially those from developing or transitioning economies [12,13]. Another study showed that quality of place factors contribute positively towards the retention of global talent [14]. 

Advanced economies are preferred by expatriates and immigrants because they can offer career advancement [5], which is especially true for those from less-developed economies [12,13,14,15]. Barrett and O’Connell suggested that individuals relocate to different countries for career advancement reasons rather than for other reasons [16]. However, the medical technology level of the medical aid sites is worse than that of China. Medical aid sites are often short of common devices, let alone the advanced devices. Access to advanced and newest devices and technology during the expatriation period is difficult, whereas medical technology updates very fast. Hence, it means CMATMs will be kept away from advanced devices and technology for two years, and they could not keep pace with the new development. When CMATMs finish the expatriation assignments and come back to China, they are behind their colleges and have to make great effort to learn new technology, which may put them at a disadvantage.

The destinations of CMATMs usually have a high prevalence of infectious diseases, such as HIV, malaria, dengue fever, and Ebola. Liang’s study on Guangxi CMATMs found that the most concerned risk was diseases, and 57% of CMATMs from Guangxi suffered from malaria infections for one to three times [17]. Overall, a high prevalence of infectious diseases and inadequate medical resources lead to high health and life risk for CMATMs. Besides, the destinations of CMATMs are with dangers from public security or even terrorist attack possibilities. The Aid Worker Security Report 2018 reported that 2017 saw a 30% rise in global fatalities compared to 2016. In 2017, 139 aid workers were killed, 102 were wounded, and 72 were kidnapped [18]. 

## 2. Materials and Methods

### 2.1. Research Design

This study adopted a qualitative descriptive approach, in which in-depth, semi-structured interviews were the primary data collection strategy [19,20,21]. Conventional content analysis was used to analyze the data. We chose qualitative description because there is a general lack of knowledge regarding influencing factors for the expatriate willingness of CMATMs. In the absence of such knowledge, quantitative methods are difficult to apply. Qualitative description allows us to “stay closer to the data and to the surface of words and events” than other qualitative traditions [20] to elicit in-depth insights from participants on their perspectives towards the themes of the present study. Ethical approval for this study was obtained from the institutional review board of School of Health Science of Wuhan University (Project Identification Code: 2016-S-0011-4/03). Data collection occurred between July 2016 and October 2016.

### 2.2. Participants Selection

Purposive sampling was used to achieve a heterogeneous selection in terms of area, institution, position, career, and architecture of these factors. In fact, Chinese medical aid team dispatching are undertaken by twenty seven provinces instead of National Health and Family Planning Commission of the People’s Republic of China, and each province has fixed recipient countries. Participants from H, B, and Q were interviewed. H has been one of the best provinces regarding medical aid team dispatching management (measured by National Health and Family Planning Commission of the People’s Republic of China). B is one of the provinces responsible for most medical aid sites, and every year it sends nearly 80 personnel to nine medical aid sites (7.9% of all the overseas medical aid sites) in Africa. Q is one of the undeveloped provinces and depends entirely on funding from central government regarding the medical aid team dispatching issue. In terms of gross domestic product (GDP), in 2014 and 2015, Provinces H was in the top five (about 35 trillion in 2014, 37 trillion in 2015), Province B was in the top ten (about 27 trillion in 2014, 29 trillion in 2015), and Province Q was in the bottom three (about 23,000 billion in 2014, 24,000 billion).

This study chose participants who are in actual charge of CMATMs dispatching. These participants were from hospitals that have sent CMATMs, and from the previous local Health and Family Planning Commission (HFPC). Participants included hospital directors and local HFPC officers who were in charge of CMATMs dispatching, and CMATMs that had returned from medical aid service. To ensure the validity of the data, participants should have different professional backgrounds and roles in work related to medical aid, and represent the architecture of personnel in medical aid sites. Hence, doctors, drivers, translators, accountants, and cooks were chosen to participate in the present study, because a medical aid site includes all these personnel.

### 2.3. Data Collection

Semi-structured interview guidelines were used to study the participants’ opinions on factors influencing expatriate willingness, and measures related to expatriate willingness. All questions were intentionally left open-ended to allow participants to describe their expatriation experiences and opinions on the topics in their own words. Interviews were conducted in the conference rooms of local HFPC.

Questions could be classified into two groups according to their purposes. In the first group of questions, participants were asked to explain their insights of influencing factors for expatriate willingness of CMATMs. Participants’ experiences and feelings were welcomed to be cited as examples. In the second group of questions, participants were asked to describe related policies that could enhance or hamper expatriation willingness to be sent as CMATMs. Then they were asked to put forward suggestions for boosting expatriation willingness of the target population in this study. Interviews lasted for 25–30 min, and were conducted until no new information was yielded. All interviews were audio-recorded and transcribed verbatim.

### 2.4. Data Analysis

Audio recordings of the interviews were transcribed within 24 h, and then reviewed against original recordings for accuracy. Inconsistency was confirmed by phoning the interviewer. Interview data were analyzed using conventional content analysis and coded. Keywords and key semantics were abstracted via an iterative process of analysis and review.

## 3. Results

### 3.1. Participants’ Characteristics

Twenty five people participated in this study. Nine were from H province, eight were from B province, and eight were from Q province. Over a half of the participants were doctors (53.8%), who had returned from medical aid service in Africa. Among these doctors, there were two team leaders from H and B, respectively. Every province sent one officer of the local HFPC and one officer of the hospital to participate in the interview. Participants included one accountant, one translator, one cook, and one driver. Officers of local HFPC and hospitals are in charge of selecting and managing CMATMs. Their duties and working experience may lead to different insights from CMATMs. Thus, they were categorized as the “A” class, including six participants. Owing to the selection criteria and managerial duty, team leaders of CMATMs also may have different insights from common CMATMs. Hence, team leaders were categorized as the “B” class, including two participants, and common CMATMs were categorized as the “C” class, including 17 participants. Characteristics of the 25 participants are presented in Table 1.

### 3.2. Influencing Factors for Expatriation Willingness

Six themes emerged during the survey and analysis, including career advancement, living conditions, personal safety, leisure, family–work conflict and doctor–patient relationship (Table 2).

Analysis did not show participants from different provinces divided in the influencing factors issue. Analysis also did not reveal that different groups of participants (officers, team leaders of CMATMs, and common CMATMs) had different opinions on this issue. However, officers cited more cases, and these cases were more representative than did the other two kinds of participants. CMATMs spent more time expressing their opinions on the influencing factors issue than the policy and suggestion issue.

#### 3.2.1. Career Advancement

Topic of career advancement took up the most time among officers and doctor CMATMs. The push and pull sides for career advancement were shed light on, although participants spent much more time talking about the pull sides, indicating that they concerned much more about the pull sides. All doctor CMATMs and officers perceived that a 2 years aid period plays a negative role in doctors’ career advancement. The negative effects have three focuses: strangeness of the original medical skills and technology, loss of chance to master new medical skills and technology, and high possibility of losing the original position. Besides old devices, medical aid sites are often in serious lack of common devices. Without the necessary medical equipment, doctors will gradually become unfamiliar with the medical technology, which they were quite skilled before being dispatched. Besides, medical skills and technology is advancing at a breathless pace. During 2 years’ expatriation, access to these new advancements is impossible, let alone mastery of them. Hence, doctors deemed that medical aid expatriation undoubtedly leads to strangeness of medical skills and technology, and offers little chance to master new skills and technology. They also worried that their original position would be replaced, because 2 years is not a short period and the hospital have to operate. Some of the doctors and officers cited some real cases of returned CMATMs having been replaced. Concern of being replaced is particularly evident among doctors in the surgical department. 

“I think the impact on the doctors in the surgery department is relatively serious. Surgery departments, such as obstetrics and gynecology, ophthalmology and surgery, have many patients and operations. So, some CMATMs from these departments found that they had already been replaced when the medical aid assignment was over. If your ability is not improved, you will be slowly marginalized and moved to a new department. This happened before.” (A-1).

“The level of medical care in medical sites is very low, and the medical equipment is so poor that it is impossible to carry out new projects. After working there for two years, your skills cannot be improved. But in China, technology is advancing by leaps and bounds. When you come back, you will find that you cannot keep up with your colleges.” (B-1).

“Going out for 2 years results in decline in business and skills. This loss is irreparable.” (C-1).

All of the doctor participants thought that technical title promotion policy could motivate CMATMs candidates. All officers respondents perceived technical title promotion policy as the most important motivation for those who are willing to assume the assignment. “Most of CMATMs who are willing are mainly motivated by the technical title promotion policy.” (A-2).

#### 3.2.2. Leisure

Leisure was mentioned by the largest number of participants, although it did not account for the most time. Due to travel regulations, CMATMs mostly stayed in the dormitory after work. However, the medical sites lacks recreational facilities and sports venue. Thus, CMATMs found the after-work life very monotonous. “They often had no other entertainment but taking a walk inside the building.” (A-4, A-6). Watching TV and surfing the Internet are the main forms of entertainment. “After-work life there is very monotonous. In addition to browsing news on internet and chatting with family, there is no other entertainment.” (C-9). However, there are few Chinese channels on TV, and the network speed is very slow. “It takes more than ten minutes to send a word document” (B-2).

Internet is considered as the core among all the entertainment means for CMATMs. Its social use (e.g., browsing social networking sites and instant messaging) and entertainment use (e.g., downloading files) were both shed light on. Participants cared about the social use of Internet more than the entertainment use of it. Chatting with family and friends via instant messenger were highlighted by participants. However, since CMATMs could only surf the Internet during free time, video chatting time with their family and friends was undoubtedly concentrated, leading to great difficulties to contact with family and friends with the network conditions in recipient countries. Lack of entertainment and difficulty of communicating with family and friends resulted in feeling lonely among CMATMs, which even led to psychological disorder and illness. “Some CMATMs even may have depression.” (A-2) and “Some tried to commit suicide.” (A-4). Moreover, consequences included disharmony and even quarrel among CMATMs. “If there was no net that day, everyone quarreled.” (C-3).

As of describing feelings or physiological state, respondents used the following words: dull, monotonous, lonely, depression, pressure, and disorder.

“The CMATMs could not go around freely, which was quite like being restricted. It easily leads to psychological distortion.” (A-4).

“They were prone to quibble during the assignment. They were depressed, and since there were no good way to release their depression. Gradually, they were prone to quarrel. A common issue, which could be solved easily in China, would be handled by them in an abnormal way. This situation is common in each batch.” (A-1).

#### 3.2.3. Living Conditions

Three themes emerged in terms of living conditions, including shortage of water and electricity, poor diet, and bad housing. Living conditions of medical sites in capital are better than those in remote regions. CMATMs concluded that life there was hard. Poor living conditions discourages expatriation willingness.

Shortage of water and electricity are the norm in many medical sites. “CMATMs often drive three kilometers away to get water.” (A-1).

“For some teams, going out of electricity and water is very common.” (C-2).

Due to the shortage of daily consumer goods production and supply, coupled with inflation, the cost of diet was high, leading to restriction of selection and consumption. “The prices were very high because of the speedy inflation.” (C-10). “Most of the local meat was beef and mutton, and it was very expensive.” (C-7). So, “We bought it once or twice a month.” (C-9). “Vegetables were usually very expensive.” (C-11) and “The variety of vegetables was very different from that in China” (C-6). Hence, many CMATMs took vegetable seeds to recipient countries and planted vegetables there. Some CMATMs also took flavorings along. Moreover, in some medical aid sites, the water was of poor quality, and harmed CMATMs’ health.

“The water quality was not good there, and the elements exceeded the standard. The hair turned white after two years. Some CMATMs had been buying mineral water. But it was very expensive if you bought mineral water for daily use……and there was no water purifier.” (C-8).

As of housing, CMATMs had to share rooms, which was of poor sound insulation effect. The household appliances were old and damaged. However, the maintenance period was very long, and the cost was very high. The recipient countries lacked spare parts for the corresponding equipment and electronic products, and sometimes they had to order from China. Besides, there was a lack of maintenance technicians. Therefore, it was difficult to repair the household appliances in time, and sometimes CMATMs had to apply for a new one, which brought inconvenience to the normal life.

“We had to share rooms. The rooms are separated by a partition, but the sound insulation effect was very poor. You could hear every sound. Household appliances were very old, slow to update, and had a long maintenance cycle.” (B-1).

#### 3.2.4. Personal Safety

Three kinds of threats to personal safety have been focused: local security, high risk of being infected with local infectious diseases, and life or health threats resulted from local inadequate medical service.

As of local security, high criminality rates, loose restriction on guns, and terrorism pose a potential threat to the personal safety of CMATMs. 

“You must return to the city before three pm. The aid site is surrounded by small hills, and anti-government forces can be seen on the road. Guns can easily be seen on the top of the hills. Although anti-government forces claimed that they would not hurt Chinese, and they had a good impression of CMATMs. But no one could prevent accidents from happening. They were all live-fired, and there were bound to have some psychological impact on CMATMs.” (A-1).

“Their guns are open and scary, which is a potential danger.” (C-15).

Owing to the prevalence of infectious diseases, CMATMs will face many more infectious patients than they do in China, which leads to higher possibility of being infected. Moreover, some infectious diseases are incurable, bringing great pressure to CMATMs. 

“CMATMs in our team used to have this psychological disorder when they came back. His gloves were broken during the operation, and he clearly knew that the patient was an AIDS patient. You can imagine his feeling. After he came back, the psychological pressure could not be erased for a long time. You can imagine how much pressure he underwent when he did the surgery.” (A-2).

“Generally, eight to ten members of every team will be infected with malaria. Although the disease can be cured there, it would relapse when they come back to China.” (A-1).

“I went there…… my hand was punctured during the operation. AIDS is very prevalent there. I was very scared at the time.” (C-17). 

Local inadequate medical service is also a risk for CMATMs if they are heavily injured or have serious illness. 

“One of our team members fell off the stairs, and his leg broke. But there was no way to treat it in the recipient site. So, we finally gave him a temporary plaster and sent him back to China. It cost seven days to come back. When the plaster was cut, the legs had turned black. Luckily, we sent him back without any hesitation. If it had cost two more days, the leg would not have been saved.” (A-4).

#### 3.2.5. Family–Work Conflict

During 2 years’ assignment, CMATMs basically could not assume family responsibilities or take care of their family. According to the regulations, CMATMs can only enjoy one family vacation paid by the government. In addition, lacking of network resources in medical sites results in much less contact with family, “It is easy to conflict with the family” (C-3).

The old and the young members in the family are mostly mentioned. “Every time I think of it, I feel I owe a lot to my parents.” (C-9).

“There are a lot of cases that the members were late for their parents’ funeral. …… Our team leader’s grandson, a five-year-old child, suddenly developed a brain tumor. The boy was going to die in hospital. Although we agreed the team leader to come back at once, he could do nothing after he came back.” (A-4).

“CMATMs suffer from career loss and family loss.” (B-1).

In sum, absence of a family role, which includes the role as spouse, parent, and offspring, negatively affects expatriation willingness to be dispatched as CMATMs.

#### 3.2.6. Doctor–Patient Relationship

Besides technician title promotion policy, doctor–patient relationship in the recipient countries can also motivate doctors to accept the expatriation. Participants who showed willingness to be present as CMATMs expressed that they would accept the assignment because they enjoyed the nice doctor–patient relationship in the recipient countries. Patients are respectful to doctors, and doctors could concentrate on offering medical service without paying extra attentions to dealing with medical disputes. Doctors have and enjoy the sense of accomplishment. 

“The doctor–patient relationship is very good there. Patients are very respectful to the doctors…. We went there again because of this specific nice relationship, but not money.” (A-6).

“Nice doctor–patient relationship is one main motivation factors for most doctors who are willing to be sent again……. You have the sense of accomplishment every time you save a patient.” (A-4).

### 3.3. Policy and Suggestions

Analysis did not show participants from different provinces divided in the policy and suggestion issue, nor revealed that different groups of participants (officers, team leaders of CMATMs, and common CMATMs) had different opinions on this issue. Topic of policy and suggestion took up much less time than topic of influencing factors. Policies and suggestions discussed were related to career advancement, entertainment, living conditions, and family–work conflict (Table 3). 

Technical title promotion policy was the core of the policy issue. All doctor respondents and officer respondents agreed that technical title promotion is the most important motivator. In terms of suggestions for career advancement, two main issues were put forward. First, adopting more effective technical title promotion policy. Second, increasing the foreign aid gross income to compensate for the 2 years’ loss.

Regarding solving leisure problems, the internet speed was the focus. “Internet is the only form of entertainment, as well as the most important form of communicating with family and friends, which could ease the psychological pressure. Hence, it is very important to solve internet problem, mainly the speed” (C-7). Increasing entertainment budget ranked the second.

As of living conditions, participants called for more investment to improve the overall conditions.

In terms of family–work conflict, participants had different opinions. Some considered shortening 2 years aid period to 1 year as an effective approach, whereas some deemed it inefficient, because “A year later, CMATMs just get adjusted and familiar with the life and work there, and have the ability do it better. But the assignment ends. This is a waste of resources.” (A-5). Some held the opinion that allowing family companionship could ease family–work conflict, while others thought it unpractical because it would not do good to the spouse’s career, or it would not be a good idea owing to the living conditions and personal safety issue.

## 4. Discussion

This study explored the expatriation willingness and its influencing factors for CMATMs. The findings showed that career advancement, leisure, living conditions, personal safety, family–work conflict, and doctor–patient relationship were related to CMATMs’ expatriation willingness. Results revealed that participants did not put forward many suggestions.

All doctor participants spent the most time talking about career advancement. This phenomenon indicates that career advancement is the most important factor and concern for doctor CMATMs. The push and pull sides for career advancement have been shed light on, although more participants concerned about the pull sides. It is understandable that these excellent doctors worry about the high possibility of falling behind in career when the expatriation ends. Local HFPC and hospitals are required to select qualified and excellent doctors from grade three hospitals as CMATMs. In another word, these selected doctors have a bright future. We could reasonably assume that these doctors value their occupational role and have high motivation for occupational achievement. Prior studies found that employees’ need for occupational achievement can have considerable influence on their willingness to accept an international assignment [13,21], because they believe the assignment would help advance their career. Previous studies have primarily suggested that employees expect to upgrade their skills, increase their knowledge, and advance their careers by relocating to advanced countries [12,13,15]. Hence, according to prior studies, employees in need of occupational achievement are willing to assume those beneficial overseas assignments. However, the present study revealed that owing to backwardness and lack of medical device, and hardly possible access to learning and mastery of new technology, assignment as CMATMs could not advance doctor CMATMs’ career, but on the contrary, have a negative impact on their career, which not only results in lagging behind in terms of knowledge and skills, but also losing the original position. For these excellent doctors, they surely take these aspects as demerits to their career advancement. It is reasonable that these doctors deem that they are likely to have greater achievement if they spend the 2 years in China instead of being dispatched as CMATMs. How to reduce the demerits to doctors’ career advancement is a core for motivating doctors, which should be the key point of policy. 

Findings of the present study suggest that enhancing technical title promotion policies is an effective approach to motivate doctors. However, we find that not all provinces benefit CMATMs in technical title promotion. In the provinces that have set a technical title promotion policy to encourage CMATMs, experience of CMATMs could not help direct technical title promotion. In some provinces, such experience could only bring priority under the same conditions. In some other provinces, such experience can exempt a man-machine oral test, which is one of the technical title promotion conditions. We suggest all provinces should benefit CMATMs in technical title promotion. Moreover, further study should explore different promotion measures.

Results demonstrate that their after-work life in the recipient country was very monotonous and dull, as well as there were too little entertainment resources. In another word, CMATMs had very poor leisure quality and experienced leisure boredom during the medical aid service period. Plenty of previous studies have proven that participating in leisure activities are beneficial to people in a numerous aspects, such as physical health, psychological, social, and economic well-being, and quality of life [22,23,24]. Furthermore, leisure is considered as an important way to reduce the negative influence of stress on physical and mental situation [22]. However, when people subjectively (lacking the ability to manage their leisure time meaningfully) or objectively (inadequate leisure resources) cannot utilize their leisure time, they probably will not receive benefits from their leisure time, resulting in leisure boredom. Leisure boredom is “a mismatch between desired arousal-producing characteristics of leisure experiences, and perceptual or actual availability of such leisure experiences” [25]. Feelings of emptiness, meaninglessness, and restlessness are considered as the characteristics of leisure boredom [26]. Literature focusing on the consequences of leisure boredom shows that boredom is related to poor mental health (e.g., loneliness, depression, and anxiety), and to problematic behavior (e.g., eating disorders and substance abuse) [27,28,29]. Leisure boredom has also been found to significantly predict low life satisfaction [26,30,31]. Leisure boredom explains depression cases, physiological disorder cases, seemingly unreasonable behavior, and proneness to quarrel among CMATMs. Results of the present study call for more resources and measures to enhance leisure life quality of CMATMs. It does not only affect the CMATMs’ expatriation willingness, but also influences their mental health, life satisfaction, and their cooperation and assignment performance. 

It is worth noticing that Internet use is the core topic for entertainment in this survey, and participants complained mostly about great difficulty of contacting with their family and friends, which was a result of the slow Internet speed. It depicts two important themes: the use of Internet, and the need to be socially connected. There are several explanations for the fact that Internet use being the core of all entertainment means in the present study. Firstly, respondents probably had been used to entertaining on the Internet, or even had relied on the Internet. As of December 2016, the number of Internet users in China reached 731 million. Data shows that in 2016, Chinese netizens spent 26.4 h a week on the Internet, roughly the same as in 2015. In 2016, the usage rate of Internet users of instant messaging, search engines, online news, online video, and online music was 91.1%, 82.4%, 74.5%, and 68.8%, respectively [32]. Secondly, loneliness, depression, and stress, which were used by participants to describe their feelings and psychological state, and which were the probable outcomes of going abroad without the companionship with family, poor leisure life, and typical characteristic of leisure boredom, would lead to CMATMs’ greater need for Internet use. Perceiving psychological problems as the cause of Internet use, previous studies have shown that psychological problems were directly related to Internet use [33,34,35]. Of all these psychological problems, loneliness attracts the most attention. Morahan’s study found that lonely individuals were more likely to use the Internet to modulate negative moods, and they used the Internet more than others [35]. Meanwhile, the benefits of Internet use should not be ignored in the present study. Shaw’s study found that Internet use could significantly increase perceived social support and self-esteem, while decrease loneliness and depression significantly [36]. Some studies also have suggested that Internet use has an overall positive effect on well-being [37,38]. Regarding the need to be socially connected, studies have shown that being socially connected is influential for psychological, emotional well-being, physical well-being [39], and overall longevity [40,41]. On the contrary, a lack of social connections is detrimental to health. Considering the two following facts: (a) participants complained mostly about great difficulty of contacting with their family and friends via Internet and (b) undoubtedly, Internet is the only and the best social connecting means for CMATMs, which not only enables CMATMs to communicate with domestic important social relationship in time, but also helps CMATMs to connect with domestic society by offering instant channels to get domestic news and comments, we could focus on probing into the social use of Internet. A 15 years’ longitudinal study showed that using Internet to communicate with offline strong ties, such as family and close friends, are beneficial to psychological well-being as measured by declines in depression, loneliness, and stress, and increases in perceived social support, mood, and life satisfaction [42]. Another study found that greater use of the Internet as a communication tool was associated with a lower level of social loneliness [43]. Overall, CMATMs subjectively have expressed their need for social use of Internet. The beneficial influence of Internet use, especially its social use, have been demonstrated by previous studies. Results of the present study and findings of previous studies are calling for the guarantee of the social use of Internet. On the other hand, we must admit that it is not easy to do so, because increasing the Internet speed is related to the network infrastructure. Making use of the Internet resource of local Chinese enterprises or other entities might be a practical approach.

Furthermore, it is interesting and noticeable that leisure was the most discussed topic among participants, rather than living conditions. It was in line with our study on CMATMs’ overseas life satisfaction, in which showed that leisure satisfaction was lower than food satisfaction and housing satisfaction [44]. These findings probably contradict to many people’s impression, which deems hard living conditions as the most concerned factor among CMATMs, because previous reports have discussed living conditions much more than entertainment. It is understandable that previous studies and media reports have taken living conditions as a more important issue, since hard living conditions are visible, instant, and more eye-catching for media coverage. However, the fact that leisure got more attention than living conditions in the present study may be due to the reason that the impact of psychological feeling lasts longer than that of living conditions, which deserves our attention. 

It is understandable that CMATMs were not satisfied with the living conditions, and the tough living conditions negatively affect their expatriation willingness, which was in line with previous studies [3,4]. Although China itself is a developing country, it is undeniable that China’s economy and living conditions have gained great achievement. China has set a high criteria for selecting CMATMs, particularly doctors. Local HFPC and hospitals are required to select qualified and excellent doctors mostly from tertiary hospitals (top level in China) as CMATMs. These hospitals often locate in relatively developed regions in China. For example, there were 67 tertiary comprehensive hospitals in Guangdong Province in 2016, and 62.9% of these hospitals locate in Guangzhou (18), Shenzhen (6), Foshan (7), Dongguan (6), Zhongshan (3), and Zhuhai (2) [45]. These six cities are the richest place in Guangdong. It is understandable that the doctors have good living conditions due to their professions and the location of the regions. However, CMATMs are often dispatched to hardship arrears of African countries, which are with hard living conditions. Africa has 33 least developed countries, 29 of which are recipients of China. Most of these recipient countries locate in Sub-Saharan Africa. 319 million people in Sub-Saharan Africa are without access to improved reliable drinking water sources [46]. Hardships mentioned by participants included shortage of water and electricity, poor diet, and bad housing, which were in line with previous studies [17,47,48]. Tian’s study revealed that some medical team sites had been out of water for three months, making daily washing a big problem [49]. While shortage of water and electricity is hard to solve by China alone, housing and diet are much easier and could be solved by China alone by investing enough funds.

Although the present study did not find that participants ranked personal safety the most important factor, personal safety is undoubtedly the most basic need of a person, and guaranteeing its citizen’s personal safety is the basic responsibility for a country and its government. Results showed that participants focused on the following three kinds of threats to personal safety: local security, high risk of being infected with local infectious diseases, and life or health threats resulted from local inadequate medical service. As of local security, participants also mentioned crime (e.g., robbery) and terrorist. These two issues fall into the scope of indicators of Ibrahim index of African governance (IIAG). IIAG measures the overall governance in all African countries and is issued annually by the Mo Ibrahim Fund. The index, which began in 2007, measures four aspects, including Safety and Rule of Law, Participation and Human Rights, Sustainable Economic Opportunity, and Human Development. The index uses a percentile system. According to IIAG, the absolute level in many categories is very low. Perception of personal safety score is 45.9 in 2013, 45.6 in 2014, 44.6 in 2015, and 42.4 in 2016 [50]. Absence of crime score is merely 47.5 in 2013, 48.4 in 2014, 48.1 in 2015, and 48.7 in 2016. In 2011, four gunmen entered the room and robbed three CMATMs. In 2012, twenty-eight Chinese workers were kidnapped in Niger [51]. Absence of domestic armed Conflict or risk of conflict, which is worried by CMATMs, is low, standing merely at 56.3 [50]. At the end of 2012, the situation in Central Africa was so volatile that the Ministry of Health had to withdraw the medical team that was carrying out the medical mission there [51]. Since non-interference in each other’s internal affairs is a principle of international law, China could only communicate with recipient country to take more measures to ensure the CMATMs’ safety. CMATMs have to face high risk of being infected as well. The prevalence of tuberculosis and the spread of HIV in sub-Saharan Africa are among the highest in the world [52]. Malaria, typhoid fever, dengue fever, and other diseases are frequent in Africa. Many media reports and studies have mentioned that some CMATMs died of illnesses after being infected with the disease, and some others suffered recurrent attacks, or could not be cured and suffered for life [53,54]. As of the threats from being infected, and from local inadequate medical service, a possible approach might be equipping medical sites with more necessary and advanced devices and medicines. 

The findings indicated that absence of taking care of family negatively affect CMATMs’ expatriation willingness, which is consistent with previous studies. Phyllis’s longitude study on 839 Australian employees demonstrated that employees with less family barriers showed stronger expatriation willingness, and the expatriation interest of childless singles were most realized [7]. Besides, a study conducted by Konopaske and his colleges suggested that eldercare also influences managerial willingness to assume global assignments [55]. The fact that participants mentioned children and elders mostly in this part suggest that policy or local government and hospitals could take measures to take care of these two populations to ease CMATMs’ worries and boost their willingness. 

The importance of family–work conflict is not only that it is one of the factors affecting expatriation willingness, but also that it has a positive correlation with depressive symptoms among employees [56,57,58,59]. Hao’s study found that family–work conflict could increase doctors’ depressive symptoms [60]. The findings of these studies call for attention for caring for mental health of CMATMs, especially offering perceived organizational support, since perceived organizational support, as a positive resource, could fight against doctors’ depressive symptoms [60].

Findings show that a nice doctor–patient relationship in recipient countries motivates CMATMs to accept a new assignment. This is in line with previous findings. A study sampling on 600 young medical staff found that doctor–patient conflicts negatively impact the retention of the participants [61]. Another study conducted among 882 medical staff showed that the doctor–patient relationship was negatively correlated to turnover intention, indicating that the better the doctor–patient relationship, the lower the turnover tendency. The study also found and doctor–patient relationship was positively correlated to job satisfaction [62]. Li’s study found that there was a positive correlation between doctor–patient relationship and doctors’ work engagement [63].

Another study conducted by us in 2016 depicted the gross aid income. Of 317 respondents, 32.2 got 130,000–190,000 RMB as the gross aid income, 30.6% got 190,000–250,000, 19.2% had less than 130,000, and 18% had more than 250,000 [44]. Economically, it is not attractive for doctors, especially excellent doctors or doctors in developed areas. Measures should be taken to increase the gross aid income.

Shortening the expatriation period may be one effective approach to reduce all the negative influences of the six factors. Since 2016, some provinces have begun to shorten the period to one year, for example, Ningxia, Guangxi, and Tianji. Of the six factors, domestic policy can have great effect on career advancement, living conditions, and leisure. In terms of career advancement, NHC can require all provinces to adopt more effective technical title promotion policy, and guarantee that returned CMATMs could have adequate and qualified technical training. Regarding living conditions, housing and diet could be solved by investing enough funds. As of leisure, recreational facilities and sports venue in the medical team site are much easier to improve than the internet resource. Since recreational facilities and a sports venue do not rely on the infrastructure or conditions of recipient countries. A basketball court, table tennis room, and movie studio are good choices. Policy should increase and ensure investment on living conditions and leisure. It is also necessary to establish minimum standards for living conditions, recreational facilities, and sports venue.

Last, a number of limitations of this study should be elucidated. First, the present study evaluated the opinions of only 25 individuals; in future studies, it might prove useful to increase the sample number to corroborate the findings of this study. Second, this is a qualitative research, hence findings of this study cannot be generalized to people who were not included in the study sample. For further study, qualitative method should be adopted.

## 5. Conclusions

This study is among the first to explore the influencing factors of expatriation willingness for CMATMs. Career advancement, leisure, living conditions, personal safety, family–work conflict, and doctor–patient relationship were found related to expatriation willingness of participants. Career advancement is the most important factor and concern for doctor CMATMs. However, all doctor CMATMs and officers perceived that a 2 years aid period plays a negative role in doctors’ career advancement. Leisure was complained by the largest number of participants. Poor leisure quality during the assignment negatively affects expatriation willingness. Slow Internet speed results in great difficulty of communicating with domestic important social relationships as well as connecting with domestic society. The tough living conditions and absence of family roles also negatively affect expatriation willingness. The technical title promotion policy is the most important and effective motivator for doctor CMTAMs. A nice doctor–patient relationship in recipient countries is the other motivator. The present study indicated that adopting more technical title promotion policy, improving living conditions and leisure facilities, shortening expatriation period, and increasing the annual aid income could be an effective way to boost expatriation willingness.

## Figures and Tables

**Table 1 ijerph-17-00632-t001:** Demographics of respondents (*n* = 25).

Province	Number	Position	Team Leader of CMATMs
H	9	1 officer of local HFPC	
		1 officer of hospital	
		7 doctors	1 team leader
B	8	1 officer of local HFPC	
		1 officer of hospital	
		3 doctors	1 team leader
		1 accountant	
		1 translator	
		1 cook	
Q	8	1 officer of local HFPC	
		1 officer of hospital	
		5 doctors	
		1 cook and driver	
Total	25		

**Table 2 ijerph-17-00632-t002:** Six influencing factors for expatriation willingness.

Categories	Subcategories
1. Career advancement	strangeness of the original medical skills and technology loss of chance to master new medical skills and technology high possibility of losing the original position technical title promotion
2. Living conditions	shortage of water and electricity poor diet bad housing
3. Personal safety	local security high risk of being infected with local infectious diseases life or health threats
4. Leisure	lack of recreational facilities and sports venue few Chinese channels slow network speed
5. Family–work conflict	much less contact with family absence of family roles
6. Doctor–patient relationship	respect from patients no need to pay extra attentions to dealing with medical disputes sense of accomplishment

**Table 3 ijerph-17-00632-t003:** Suggestions for motivating CMATMs candidates.

Categories	Subcategories (Descending Order)
1. Career advancement	adopting more effective technical title promotion policy increasing the foreign aid gross income
2. Living conditions	more investment
3. Leisure	improving the internet speed increasing entertainment budget
4. family–work conflict	shortening aid period allowing family companionship

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
