# Peer review of "Influencing Factors for Expatriation Willingness of Chinese Medical Aid Team Members (CMATMs) in Africa: A Qualitative Descriptive Study"

_ijerph, 2020, doi:10.3390/ijerph17020632_

Round 1

Reviewer 1 Report

The article presents an interesting issue of the work of medical personnel on foreign missions. The article is clear and careful. The selection of research hypotheses is well supported by literature and proved. One can get the impression that the article strongly emphasized the negative effects of being CMATMs. It is difficult for me to indicate what would convince me to become CMATMs. From the article it can be concluded that working as a CMATMs is just a series of sacrifices. Is it true? Who decides to become CMATMs and why? What are the perks of this? This is not clear to me. I don't see any benefit from taking the job as CMATMs. What are the benefits of working as CMATMs? Data visualization is very important. It increases the attractiveness of the article and makes it easier to read. I strongly recommend extending the article with a map or overview graphics of the research locations (provinces B, Q and H). This will increase the attractiveness of the article. Why was no statistical analysis performed? Statistical analysis could lead to new conclusions. In selected places the text is illegible – the numbers blend together. It's hard to read – Line 462-464. Maybe a table would be a better option. There is a big disproportion between the discussion and the summary.

Author Response

Dear Editors and Reviewers,

We very much appreciate the thoughtful and critical feedback from the reviewers. We are delighted at your decision that we revise and resubmit this manuscript. The reviewers have requested some further clarification and revisions to our manuscript, which we have completed, and all changes have been highlighted for ready identification. In addition, we have addressed each of the comments from you specifically, and the following points are the replies to editor and each reviewer point by point.

* Please note that the page and line number may be changed after MS is changed.

We are looking forward to hearing from you soon.

Sincerely,

Authors

Response to Reviewer 1 Comments

Point 1: From the article it can be concluded that working as a CMATMs is just a series of sacrifices. Is it true?

Response 1: During 1960-1980, many doctors were willing to be CMATMs, because the perks were better than the income they earned in China, and the experience of being CMATMs was helpful for getting promoted. During that time, being a CMATMs brings political benefit as well. But since 1990s, doctors’ income has exceeded the perks of CMATMs, and some promotion policies have been cancelled. It is true that fewer and fewer doctors are willing to become CMATMs. Indeed, there are some willing doctors nowadays. Some like Africa and are willing to be CMATMs, and some simply what to help African people. Such spirits are respectable. But the work of dispatching CMATMs cannot rely on respectable doctors’ spirit—it cannot ensure the long lasting and big scale selection. Encouraging and effective policy is the key to ensure the selection, as well as the job performance of CMATMs.

Point 2: Who decides to become CMATMs and why? Is it a sacrifice?

Response 2: As discussed in our paper, CMATMs have a significant impact on the implementation of China’s health assistance strategies in Africa. Hence to ensure the effect, selecting CMATMs is basically a bureaucratic choice, although the government encourages volunteers to become CMATMs. Local (Provincial) Health and Family Planning Commission (HFPC) is responsible for selecting CMATMs. Qualified hospitals (tertiary hospitals) are required to select qualified doctors. HFPC reviews the selection, organizes training and administers the medical aid team.

Point 3: What are the perks of becoming CMATMs? What are the benefits of working as CMATMs? Data visualization is very important. It increases the attractiveness of the article and makes it easier to read.

Response 3:

Nowadays, the economical perks are not attractive, and the benefits of working as CMATMs are mainly about technical title promotion policy. But not all provinces benefit CMATMs in technical title promotion. Some provinces have set technical title promotion policy to encourage CMATMs. Such experience could not help direct technical title promotion, but it could bring priority under the same conditions. In some provinces, such experience can exempt a man-machine oral test, which is one of the technical title promotion conditions.

Since technical title promotion policy is the most important motivator, we have discussed it further. Line 392-398.

We have also added the perks data in the MS. Line 549-553. Another study conducted by us in 2016 depicted the gross aid income. Of 317 respondents, 32.2 got 130,000~190,000 RMB as the gross aid income, 30.6% got 190,000~250,000, 19.2% had less than 130,000, and 18% had more than 250,000. Economically, it is not attractive for doctors, especially excellent doctors or doctors in developed areas.

We appreciate the suggestion--Data visualization.

Point 4: I strongly recommend extending the article with a map or overview graphics of the research locations (provinces B, Q and H).

Response 4: Thanks for the suggestion. We have given more details. Line 131-134.

In terms of Gross Domestic Product(GDP), in 2014 and 2015, Provinces H was in the top five (about 35 trillion in 2014, 37 trillion in 2015), Province B was in the top ten(about 27 trillion in 2014, 29 trillion in 2015), and Province Q was in the bottom 3(about 23000 billion in 2014, 24000 billion).

Point 5: Why was no statistical analysis performed?

Response 5: Very good suggestion. We will do so in the future. We have mentioned this point for future inquiries in the Limitation. Line 571-573.

In this qualitative study, we used Conventional content analysis (CCA). CCA is usually used in studies that focus on exploring a phenomenon when the existence of theoretical literature or research on the phenomenon is limited. In this way, researchers use inductive logic, avoiding imposing any categorical system of data interpretation. In addition, the CCA offers researchers the ability to give their findings theoretical cohesion, which may potentially inform the direction of future research and substantiate future findings.

Point 6: In selected places the text is illegible – the numbers blend together.

Response 6: We have revised and cited some examples. Line 475-479.

Local HFPC and hospitals are required to select qualified and excellent doctors mostly from tertiary hospitals (top level in China) as CMATMs. These hospitals often locate in relatively developed regions in China. For example, there were 67 tertiary comprehensive hospitals in Guangdong Province in 2016, and 62.9% of these hospitals locate in Guangzhou (18), Shenzhen (6), Foshan (7), Dongguan (6), Zhongshan (3) and Zhuhai (2)[45]. These six cities are the richest place in Guangdong.

Point 7: It's hard to read – Line 462-464. Maybe a table would be a better option.

Response 7: Yes, table make points clearer. Thanks for the suggestion. We have made a table of 6 influencing factors in the result section, including the subcategories (Table 2, Line 188).  Besides, we have also made a table of suggestions (Table 3, Line 343). Valuable suggestion!

Point 8: There is a big disproportion between the discussion and the summary. 

Response 8: We have revised the conclusion. Line 579-591.

This study is among the first to explore the influencing factors of expatriation willingness for CMATMs. Career advancement, leisure, living conditions, personal safety, family-work conflict, and doctor-patient relationship were found related to expatriation willingness of participants. Career advancement is the most important factor and concern for doctor CMATMs. However, all doctor CMATMs and officers perceived that 2 years’ aid period plays a negative role in doctors’ career advancement. Leisure was complained by the largest number of participants. Enhancing technical title promotion policies is an effective approach to motivate doctors. Poor leisure quality during the assignment negatively affect expatriation willingness. Slow Internet speed results in great difficulty of communicating with domestic important social relationship as well as connecting with domestic society. The tough living conditions and absence of family roles also negatively affect expatriation willingness. Social use of Internet is on the core of entertainment. Technical title promotion policy is the most important and effective motivator for doctor CMTAMs. Nice doctor-patient relationship in recipient countries is the other motivator. The present study indicated that adopting more technical title promotion policy, improving living conditions and leisure facilities, shortening expatriation period, and increasing the annual aid income could be an effective way to boost expatriation willingness.

Reviewer 2 Report

REVIEW

ID: IJERPH-691461

Article: “Influencing Factors for Expatriation Willingness of Chinese Medical Aid Team Members (CMATMs) in Africa: A Qualitative Descriptive Study.”

Abstract: The study analyses factors affecting CMATM participation. In 25 interviews, six factors are found to be of critical importance.

General Impression: This is an interesting article on an important topic.

Line:

N.B., I did not correct the English language. A good copy editor should be employed to do that.

31-34. The first sentence is too long.

44-45. How else could it be? Delete sentence.

46-47. Self-explanatory. Delete sentence.

Move hypotheses to the front of the arguments. Why the four hypotheses only. Literature review later reveals more than four, such as patient-doctor relationship. Alternatives: drop the four specific hypotheses or add a few more from previous literature. Keep the argumentation as is. Purposive sampling. Explain.

132-133. Why would the hospital directors know better than the program participants?

Conventional content analysis. Explain. What are captains? Explain. CMATM vs. CMTAM? Numbers don’t match categories.

178-179. Topic of… sentence. I don’t understand.

197- Good use of citations.

What was the verdict on hypotheses 1 & 2? … did not probe much into policy. How do you separate policy from complaints? To me they seem the same thing. …push and pull… Explain. How are the CMATM members chosen? Is it a free or a bureaucratic choice? Descriptive statistics for destination countries?

399-402. Good use of internet statistics.

461-470. Explain Ibrahim index. Please note that the index changes from year to year are not statistically significant. What one can say is that the absolute level in many categories is quite low.

488-490. Great similarity to the literature on the hardships of diplomatic life.

498-500. I don’t understand the sentence.

505-510. Be explicit which of the six factors can be affected by policies. I think the answer is more than two.

Thanks for the interesting paper!

Author Response

Dear Editors and Reviewers,

We very much appreciate the thoughtful and critical feedback from the reviewers. We are delighted at your decision that we revise and resubmit this manuscript. The reviewers have requested some further clarification and revisions to our manuscript, which we have completed, and all changes have been highlighted for ready identification. In addition, we have addressed each of the comments from you specifically, and the following points are the replies to editor and each reviewer point by point.

* Please note that the page and line number may be changed after MS is changed.

We are looking forward to hearing from you soon.

Sincerely,

Authors

Response to Reviewer 2 Comments

Regarding the English language, we will employ an editor if our MS is accepted.

Point 1: 31-34. The first sentence is too long.

Response 1: We have revised the sentence.

Point 2: 44-45. How else could it be? Delete sentence.

Response 2: We have deleted the sentence.

Point 3: 46-47. Self-explanatory. Delete sentence.

Response 3: We have deleted the sentence.

Point 4: Alternatives: drop the four specific hypotheses or add a few more from previous literature.

Response 4: We have dropped the four specific hypotheses.

Point 5: Purposive sampling. Explain.

Response 5: A purposive sample is a non-probability sample that is selected based on characteristics of a population and the objective of the study. It is an important sampling method in qualitative research. For example, in the qualitative study of grounded theory, the acquisition of research data is not through random sampling, but based on the richness of the data. For example, we have to study the reasons for the wrong drug delivery, it is best to interview the pharmacist who has given the wrong drug, rather than from the drug bureau randomly selected several pharmacists to do interviews.

Point 6: 132-133. Why would the hospital directors know better than the program participants?

Response 6: Because during the whole selecting procedure, they will talk with CMATMs candidates and persuade them (candidates who refused the assignment would talk with the directors). They knew more information—maybe different candidates refused the assignment for different reasons. 

Point 7: Conventional content analysis. Explain.

Response 7: Conventional content analysis (CCA) is usually used in studies that focus on exploring a phenomenon when the existence of theoretical literature or research on the phenomenon is limited. In this way, researchers use inductive logic, avoiding imposing any categorical system of data interpretation. In addition, the CCA offers researchers the ability to give their findings theoretical cohesion, which may potentially inform the direction of future research and substantiate future findings. Data analysis starts with reading all data repeatedly to achieve immersion and obtain a sense of the whole as one would read a novel. Then, data are read word by word to derive codes by first highlighting the exact words from the text that appear to capture key thoughts or concepts. Next, the researcher approaches the text by making notes of his or her first impressions, thoughts, and initial analysis. As this process continues,

labels for codes emerge that are reflective of more than one key thought. These often

come directly from the text and are then become the initial coding scheme. Codes

then are sorted into categories based on how different codes are related and linked.

These emergent categories are used to organize and group codes into meaningful

clusters.

Point 8: What are captains? Explain.

Response 8:  Captains are medical team leaders. We have revised.

Point 9: CMATM vs. CMTAM?

Response 9: Sorry for the mistake. CMATMs are correct. We have revised.

Point 10: Numbers don’t match categories.

Response 10: Sorry for the mistake. We have revised. Line 176.

Point 11: 178-179. Topic of… sentence. I don’t understand.

Response 11: We wanted to say “CMATMs spent more time expressing their opinions on the influencing factors issue than the policy and suggestion issue.” To make it clearer, we have revised the sentence. Line 186-187.

Point 12: What was the verdict on hypotheses 1 & 2?

Response 12: We have dropped the 4 hypotheses (See response 4). But we have made a summary for the family-work conflict (Line 322-323.) and living conditions (Line 254-255) in the result. In sum, absence of family role, which includes the role as spouse, parent and offspring, negatively affects expatriation willingness to be dispatched as CMATMs. Poor living conditions discourages expatriation willingness.

Point 13: did not probe much into policy. How do you separate policy from complaints? To me they seem the same thing.

Response 13: Thanks for the suggestion. We have revised. Line 367.

Point 14: push and pull… Explain

Response 14:

push factor: something that makes people want to leave a place or escape from a particular situation.

pull factor-:something that attracts people to a place or an activity.

Point 15: How are the CMATM members chosen? Is it a free or a bureaucratic choice?

Response 15: Selecting CMATMs is basically a bureaucratic choice, although the government encourages volunteers to become CMATMs. Local (Provincial) Health and Family Planning Commission (HFPC) is responsible for selecting CMATMs. Qualified hospitals are required to select qualified doctors. HFPC reviews the selection, organizes training and administers the medical aid team.

Point 16: Descriptive statistics for destination countries?

Response 16: Thanks for the suggestion. We have added some descriptive statistics. Line 482-484, 486-487,515-519.

Point 17: 461-470. Explain Ibrahim index. Please note that the index changes from year to year are not statistically significant. What one can say is that the absolute level in many categories is quite low.

Response 17: We have explained Ibrahim index in the MS, cancelled some sores and added some examples. Line 497-511.

The Ibrahim African Governance Index measures the Overall Governance in all African countries and is issued annually by the Mo Ibrahim Fund. The index, which began in 2007, measures four aspects, including Safety & Rule of Law, Participation & Human Rights, Sustainable Economic Opportunity, and Human Development. The index uses a percentile system. 

Point 18: 488-490. Great similarity to the literature on the hardships of diplomatic life.

Response 18: We have cancelled the sentence.

Point 19: 498-500. I don’t understand the sentence.

Response 19: We have revised the sentences. 542-549.

Here are the revised sentences “Another study conducted among 882 medical staff showed that doctor-patient relationship is negatively correlated to turnover intention, indicating that the better the doctor-patient relationship, the lower the turnover tendency. The study also found and doctor-patient relationship is positively correlated to job satisfaction. Li’s study found that there was a positive correlation between doctor-patient relationship and doctors’ work engagement.”

Point 20: 505-510. Be explicit which of the six factors can be affected by policies. I think the answer is more than two.

Response 20: Thanks for the valuable suggestion. It makes the findings and discussions more valuable. Line 555-566.